# The Upregulation of Leucine-Rich Repeat Containing 1 Expression Activates Hepatic Stellate Cells and Promotes Liver Fibrosis by Stabilizing Phosphorylated Smad2/3

**DOI:** 10.3390/ijms25052735

**Published:** 2024-02-27

**Authors:** Yake Wang, Xiaolong Li, Xiaowen Guan, Zhe Song, Huanfei Liu, Zhenzhen Guan, Jianwei Wang, Lina Zhu, Di Zhang, Liang Zhao, Peitong Xie, Xiaoyi Wei, Ning Shang, Ying Liu, Zhongzhen Jin, Zhili Ji, Guifu Dai

**Affiliations:** School of Life Sciences, Zhengzhou University, Zhengzhou 450001, China; mswyk1314@163.com (Y.W.); 18226173047@163.com (X.L.); songzhe@zzu.edu.cn (Z.S.); huanfeiliu@163.com (H.L.); 15993478401@163.com (Z.G.); wjw725@126.com (J.W.); 13588819821@163.com (L.Z.); diudiuzang@163.com (D.Z.); zl875485141@163.com (L.Z.); xptlyrj@163.com (P.X.); weiyi314159@163.com (X.W.); 202011161010127@gs.zzu.edu.cn (N.S.); liuyingzlyaa@163.com (Y.L.); jzhongzhen@163.com (Z.J.); 13271386805@163.com (Z.J.)

**Keywords:** liver fibrosis, LRRC1, p-Smad2/3, HSC activation, ubiquitination

## Abstract

Liver fibrosis poses a significant global health risk due to its association with hepatocellular carcinoma (HCC) and the lack of effective treatments. Thus, the need to discover additional novel therapeutic targets to attenuate liver diseases is urgent. Leucine-rich repeat containing 1 (LRRC1) reportedly promotes HCC development. Previously, we found that LRRC1 was significantly upregulated in rat fibrotic liver according to the transcriptome sequencing data. Herein, in the current work, we aimed to explore the role of LRRC1 in liver fibrosis and the underlying mechanisms involved. LRRC1 expression was positively correlated with liver fibrosis severity and significantly elevated in both human and murine fibrotic liver tissues. LRRC1 knockdown or overexpression inhibited or enhanced the proliferation, migration, and expression of fibrogenic genes in the human hepatic stellate cell line LX-2. More importantly, LRRC1 inhibition in vivo significantly alleviated CCl_4_-induced liver fibrosis by reducing collagen accumulation and hepatic stellate cells’ (HSCs) activation in mice. Mechanistically, LRRC1 promoted HSC activation and liver fibrogenesis by preventing the ubiquitin-mediated degradation of phosphorylated mothers against decapentaplegic homolog (Smad) 2/3 (p-Smad2/3), thereby activating the TGF-β1/Smad pathway. Collectively, these results clarify a novel role for LRRC1 as a regulator of liver fibrosis and indicate that LRRC1 is a promising target for antifibrotic therapies.

## 1. Introduction

Liver fibrosis is characterized by the excessive accumulation of extracellular matrix (ECM) in the liver, and it can be considered the consequence of prolonged hepatic injury caused by various factors, including hepatitis virus infection, alcohol misuse, and nonalcoholic steatohepatitis [1]. If uncontrolled, liver fibrosis gradually progresses to liver cirrhosis and even hepatocellular carcinoma (HCC), ultimately leading to organ dysfunction and mortality and seriously threatening human health [2,3,4]. Unfortunately, there is still a lack of effective medical therapies for the treatment of liver fibrosis [5,6,7]. Thus, an in-depth understanding of the molecular basis of liver fibrosis is important and necessary for supporting the development of novel antifibrotic therapies.

Hepatic stellate cells (HSCs) located in the space of Disse play pivotal roles in the occurrence and progression of liver fibrosis [8,9]. Under normal conditions, HSCs typically remain in a quiescent state [10]. However, upon liver injury or exposure to specific microenvironmental cues, HSCs undergo a process of activation, depleting their retinoid reserves and transforming into myofibroblast-like cells that express α-smooth muscle actin (α-SMA) [1]; the activation of quiescent HSCs could be triggered and perpetuated via autocrine/paracrine mechanisms, resulting in the secretion and accumulation of ECM components, such as collagen and laminin [5,10]. In particular, damaged hepatocytes, endothelial cells, and activated Kupffer cells initiate and perpetuate HSC activation by secreting growth factors and proinflammatory cytokines, including interleukin-1β (IL-1β) [11,12], platelet-derived growth factor (PDGF) [13], and transforming growth factor-β1 (TGF-β1) [14,15]. Together, these factors and cytokines create a profibrotic microenvironment to accelerate the development of liver fibrosis.

As one of the most important profibrogenic cytokines involved in the activation of HSCs, TGF-β1 interacts with its receptors, triggering the survival, differentiation, and activation of myofibroblasts through both canonical and noncanonical signaling pathways [16,17,18,19,20]. The canonical pathway, which involves the mothers against decapentaplegic homolog (Smad)-mediated signaling cascade, has profound effects on various pathological processes. Following the activation of TGF-β1 receptors, phosphorylated Smad2/3 (p-Smad2/3) translocates to the nucleus from the cytoplasm, governing the transcription of fibrogenic genes [16,19,20]. The stability and activity of TGF-β1 signaling components are reportedly significantly influenced by posttranslational modifications, especially ubiquitination [21,22,23,24]. NEDD4L, an E3 ubiquitin ligase, has been recognized as the primary ubiquitin ligase for the degradation of p-Smad2/3 [25,26]. Furthermore, Smad7 can promote the ubiquitin ligase Smurf2-mediated degradation of TGF-β1 receptors to inhibit TGF-β1 signaling [27].

Leucine-rich repeat containing 1 (LRRC1), also referred to as LANO, belongs to the leucine-rich repeat and PDZ domain (LAP) family [28]. LRRC1 was initially discovered to regulate cell polarity and was subsequently revealed to have diverse functions in various cell types. It is implicated in numerous biological processes, encompassing cell proliferation, stemness, and differentiation [29,30,31,32,33]. In addition, abnormal expression of LRRC1 has been observed in various types of tumors, such as HCC [29], lung cancer [32], breast cancer, and so on [33]. Notably, in our previous study, we observed the significant upregulation of LRRC1 expression in the transcriptome sequencing data of CCl_4_-induced fibrotic rat liver samples (the transcriptome sequencing was entrusted to Beijing Berry Hekang Biotechnology Co., Ltd., Beijing, China, project No. ABFC20201050; the average fpkm value of LRRC1 in the Oil group was 0.21, and that in the CCl_4_ group was 0.58; *p* < 0.05, *n* = 3). However, whether LRRC1 is related to the process of liver fibrosis and the potential underlying mechanisms have not been determined.

Therefore, in the present work, we sought to explore the essential role of LRRC1 in the progression of liver fibrosis and the underlying mechanisms involved. LRRC1 expression was significantly elevated in activated HSCs and fibrotic liver tissues from humans and mice. Moreover, in vitro investigations demonstrated that LRRC1 accumulation enhanced the proliferation, migration, and expression of fibrogenic genes in LX-2 cells. Notably, the attenuation of LRRC1 in vivo relieved liver fibrogenesis in CCl_4_-induced fibrotic mice. Furthermore, LRRC1 promoted HSC activation and liver fibrosis mainly by enhancing the stability of p-Smad2/3 via ubiquitination, thereby contributing to the activity of the TGF-β1/Smad pathway. These findings offer direct experimental evidence that LRRC1 has a crucial function in liver fibrosis and may be a potential target for therapeutic interventions against liver fibrosis.

## 2. Results

### 2.1. LRRC1 Is Highly Expressed in Human and Murine Liver Fibrotic Tissues

To explore the potential correlation between liver fibrosis and LRRC1 expression, we first analyzed the transcription level of LRRC1 using bulk gene expression data sourced from the National Cancer for Biotechnology Information Gene Expression Omnibus (GEO) database. As depicted in Figure 1A, the mRNA expression of LRRC1 was significantly increased (*p* < 0.001) in liver tissue from patients with hepatitis B virus (HBV) infection (GSE38941), alcohol abuse (GSE28619), or hepatitis C virus (HCV) infection (GSE6764) than in normal liver tissues. Additionally, there was a positive correlation between LRRC1 mRNA expression and the severity of liver fibrosis in patients with HBV infection (GSE84044; Spearman correlation analysis, R = 0.345, *p* < 0.001) and alcohol abuse (GSE103580; Spearman correlation analysis, R = 0.347, *p* < 0.001), as shown in Figure 1B,C. Then, we evaluated the protein levels of LRRC1 in human fibrotic livers. Consistent with the mRNA expression data, the protein levels of LRRC1 in fibrotic livers were significantly higher than those in normal livers, as shown by the immunohistochemistry (IHC) results (Figure 1D–F). Furthermore, two murine liver fibrosis models were used to evaluate the associations between the expression of LRRC1 and ECM genes as well as α-SMA, the definitive marker of activated HSCs. According to the HE and Sirius Red staining results, severe inflammatory cell infiltration, remarkable fibrotic progression, and elevated levels of LRRC1 and α-SMA were observed in fibrotic liver tissues from both CCl_4_-induced rats and bile duct ligation (BDL)-induced mice (Figure 2A–D). Moreover, the mRNA and protein levels of Collagen I (Col-I) in the livers of the model animals were markedly greater than those in the controls, which was in accordance with the changes in α-SMA and LRRC1 expression (*p* < 0.01; Figure 2E,F). These results suggested that the expression of LRRC1 was elevated in both human and murine fibrotic liver tissues and positively associated with the extent of liver fibrosis.

### 2.2. LRRC1 Promotes the Activation of HSCs

Several studies have shown that the activation of HSCs is a vital factor for the progression of liver fibrosis [1,6,10]. To assess whether LRRC1 contributes to HSC activation in liver fibrosis, human HSCs (LX-2) and isolated rat primary HSCs (pHSCs) were separately stimulated with TGF-β1 (5 ng/mL) for 12 h, 24 h, or 48 h to induce activation, and the protein expression of LRRC1 was observed to gradually increase from 0 h to 48 h in both LX-2 cells and pHSCs, a finding that was consistent with the increasing trend of protein expression of Col-I, Collagen III (Col-III), and α-SMA (Figure 3A,B), which revealed that LRRC1 was a possible TGF-β1-responsive gene. Moreover, to explore the effects of LRRC1 on the proliferation and migration of LX-2 cells, we transfected LRRC1-knockdown (shLRRC1) or LRRC1-overexpressing (OE-LRRC1) plasmids into LX-2 cells to inhibit or enhance the expression of LRRC1, respectively, and the efficiency of shLRRC1 or OE-LRRC1 was verified through qRT-PCR and Western blot analysis (Figure 3C–E). The MTT assay results showed that inhibiting or overexpressing LRRC1 reduced or promoted the proliferation of LX-2 cells, respectively (Figure 3F,G). The flow cytometry and Western blot results indicated that LRRC1 suppression led to an increase in the number of cells in the G2/M phase and a reduction in cells in the S phase, accompanied by a decrease in the protein expression of cyclin B1 and CDK1, which are related to cell cycle regulation. However, the overexpression of LRRC1 had opposite effects on the distribution of cells in different cell cycle phases and the expression of cell cycle regulatory proteins (Appendix A). This finding indicated that LRRC1 can influence cell proliferation by impacting the cell cycle distribution of LX-2 cells. Cell migration plays a crucial role in various biological processes. During early liver fibrosis, activated HSCs migrate to the site of liver injury, where they engage in ECM synthesis and release and then participate in tissue repair and fibrosis formation. Therefore, the migratory ability of HSCs is an important characteristic of HSC activation, and it is essential for the progression of liver fibrosis. In our research, we conducted a Transwell assay to investigate the impact of LRRC1 on the migration ability of LX-2 cells. As expected, LRRC1 knockdown markedly suppressed the migration ability of LX-2 cells, while LRRC1 overexpression promoted cell migration (Figure 3H,I). Moreover, the protein levels of α-SMA, Col-I, and Col-III in LRRC1-knockdown cells and in LRRC1-overexpressing cells were significantly decreased or increased, respectively (Figure 3J,K). Therefore, LRRC1 is likely a TGF-β1-responsive gene that can promote HSC activation.

### 2.3. LRRC1 Regulates HSC Activation via the TGF-β1/Smad Pathway

Recent studies have indicated that the TGF-β1/Smad pathway is essential and crucial for HSC activation and liver fibrosis [16,17,18,19,20]. Thus, to verify whether LRRC1 regulates the TGF-β1/Smad pathway, we assessed the protein expression of TGF-β1/Smad-pathway-related genes including TGFβRI, TGFβRII, Smad2/3, Smad7, and the phosphorylated TGFβRII (p-TGFβRII) and Smad2/3 (p-Smad2/3) in LRRC1-knockdown or LRRC1-overexpressing LX-2 cells through Western blotting. As shown in Figure 4A–C, among the abovementioned proteins, only p-Smad2/3 was significantly decreased in the LRRC1-knockdown cells and increased in the LRRC1-overexpressing cells. Furthermore, we found that LRRC1 knockdown attenuated the enhanced migration ability and elevated the levels of p-Smad2/3, α-SMA, Col-I, and Col-III in LX-2 cells stimulated with TGF-β1 by performing Transwell and Western blot assays (Figure 4D–G). In addition, compared with those in TGF-β1-stimulated cells, the protein levels of p-TGFβRII, p-Smad2/3, α-SMA, Col-I, and Col-III were significantly lower in cells treated with LY2109761 (20 µM), a dual-function inhibitor of TGFβRI and TGFβRII, whereas LRRC1 overexpression partially counteracted the inhibitory effect of LY2109761 (Figure 4H,I). Overall, these results indicated that LRRC1 plays an important role in promoting the activation of HSCs by regulating the TGF-β1/Smad pathway.

### 2.4. Knockdown of LRRC1 Alleviates CCl_4_-Induced Liver Fibrosis

To confirm whether targeting LRRC1 could suppress the progression of liver fibrosis in vivo, an adeno-associated virus (AAV)–shLRRC1 vector was injected into CCl_4_-induced mice via a high-pressure hydrodynamic method. AAV–shLRRC1 and empty AAV vectors did not cause liver damage after injection into the mice for more than 4 weeks (Appendix A). The HE, Sirius Red, and IHC staining results demonstrated that LRRC1 inhibition through AAV–shLRRC1 effectively alleviated CCl_4_-induced liver fibrosis, as evidenced by lower Ishak scores, a reduced Sirius-Red-stained area, and decreased α-SMA IHC scores compared to those in the CCl_4_+AAV–empty group (Figure 5B–F). In addition, the hepatic content of hydroxyproline (Hyp) and the serum levels of AST, ALT, and AKP were also significantly decreased in mice injected with AAV–shLRRC1 compared to those in the CCl_4_+AAV–empty group (Figure 5G–J). In addition, qRT-PCR and Western blot analysis of the expression levels of fibrogenic genes, including α-SMA, Col-I, and Col-III, demonstrated that the mRNA and protein levels of these genes were significantly decreased in LRRC1-knockdown mice compared to those in the CCl_4_+AAV–empty group (Figure 6A,B). Importantly, the inhibition of LRRC1 also resulted in a reduction in the protein expression of p-Smad2/3, as determined through Western blot and IHC analyses (Figure 6C–F), which are consistent with the results of studies in vitro. In conclusion, these results collectively suggested that LRRC1 plays an important role in promoting hepatic fibrogenesis.

### 2.5. LRRC1 Expression Inhibited p-Smad2/3 Proteasomal Degradation

Ubiquitination has been reported to play a crucial role in modulating the stability and activity of TGF-β1 signaling components, including p-Smad2/3 [21,23]. Thus, we speculated that LRRC1 may regulate the TGF-β1/Smad pathway via ubiquitination. Western blot analysis indicated that the protein level of p-Smad2/3 was decreased in LRRC1-knockdown cells, while the treatment of MG132 blocked this effect, suggesting that LRRC1 might affect the stability of the p-Smad2/3 protein through the ubiquitin-proteasome pathway (Figure 7A,B). Notably, recent studies have demonstrated that NEDD4L, an E3 ubiquitin ligase, has been identified as the primary ubiquitin ligase responsible for selectively targeting activated Smad2/3 for destruction, thereby influencing its stability [25,26]. We further examined the relationship between NEDD4L and LRRC1. The results showed that the protein expression of NEDD4L was not altered in LRRC1-knockdown cells and that the protein level of LRRC1 did not significantly change after NEDD4L was knocked down by siRNA (Figure 7C). However, the knockdown of NEDD4L blocked the inhibitory effect of shLRRC1 on the protein level of p-Smad2/3 (Figure 7C), suggesting that NEDD4L is necessary for LRRC1-mediated regulation of the protein level of p-Smad2/3. Furthermore, immunoprecipitation (IP) analysis revealed that the knockdown of LRRC1 enhanced the ubiquitin levels of NEDD4L and p-Smad2/3, whereas the overexpression of LRRC1 attenuated this phenomenon (Figure 7D–E); moreover, the total ubiquitin level increased or decreased when LRRC1 was inhibited or overexpressed, respectively, in LX-2 cells (Figure 7E), suggesting that LRRC1 may impede the interaction between NEDD4L and p-Smad2/3, thereby hindering the ubiquitin-mediated degradation of p-Smad2/3. In summary, these results demonstrated that LRRC1 regulated the TGF-β1/Smad pathway by inhibiting p-Smad2/3 proteasomal degradation via ubiquitination.

## 3. Discussion

Liver fibrosis serves as a crucial pathological stage in the progression of liver cancer. Despite the promising outcomes observed in preclinical investigations, the effectiveness of antifibrotic candidates in clinical trials is limited [5,6,7,34]. Thus, more potential targets for the development of anti-liver-fibrosis therapeutics with thoroughly explained molecular mechanisms are urgently needed.

Recently, LRRC1, a cytoplasmic scaffold protein, was identified as an oncogenic factor that contributes to HCC [29,35] and lung cancer [32,36]; other studies have reported that LRRC1 regulates the stemness of breast cancer stem cells to behave as a potential tumor suppressor and that in human mesenchymal stem cells (MSCs) [33], LRRC1 acts as a downstream target of PPARγ that regulates adipocyte differentiation [30]. The results of these studies together emphasize the complex roles of LRRC1 in various pathological processes. However, the specific functions and mechanisms of LRRC1 in the progression of liver fibrosis have never been reported. In the current study, we found that LRRC1 was upregulated in human and murine fibrotic liver tissues and positively associated with the severity of liver fibrosis. In addition, we also provided evidence showing that LRRC1 regulates HSC activation by stabilizing phospho-Smad2/3, thereby promoting the development of liver fibrosis. More importantly, our data showed that inhibiting LRRC1 expression in vivo can alleviate liver fibrosis in a CCl_4_-induced mouse model, implying that LRRC1 could serve as a promising target for treating liver fibrosis.

HSC activation has been characterized as a vital factor for promoting liver fibrosis. Activated HSCs exhibit typical properties, including proliferation, migration, and enhanced ECM deposition, and are accompanied by changes in gene expression [8,9,10]. In our study, we found that the knockdown of LRRC1 inhibited the proliferation, migration, and expression of fibrogenic genes, such as α-SMA, Col-I, and Col-III, in LX-2 cells, indicating that LRRC1 has a regulatory function in the activation of HSCs. As we all know, TGF-β1 signaling is a critical and important driver of HSC activation. Recent studies have demonstrated that Scribble, the paralog of LRRC1, is involved in the TGF-β1-activated pathway [37,38] and that another member of the LAP family, Erbin, negatively regulates TGF-β1-induced EMT via the ERK signaling pathway [37,38,39,40]. These conflicting findings indicate that LAP family proteins are related to TGF-β1 signaling. However, no direct evidence has been provided for the function of LRRC1 in mediating TGF-β1 signaling. Our study revealed that LRRC1 may be a TGF-β1-responsive gene. The expression of LRRC1 was increased in both LX-2 cells and pHSCs stimulated with TGF-β1, and we discovered that suppressing LRRC1 hindered TGF-β1-induced p-Smad2/3 expression and migration in LX-2 cells.

The phosphorylation of Smad2/3 is the key step for the TGF-β1 canonical signaling pathway. In addition to TGF-β1, p-Smad2/3 can be regulated by several other factors; for example, Yu J.S. et al. reported that p-Smad2/3 could be reduced through the inhibition of the PI3K/mTORC2 pathway [41]; SIRT1 was reported to interact with Smad2/3 to impede p-Smad2/3 nuclear translocation [42]; and NEDD4L, an E3 ubiquitin ligase, was shown to bind to p-Smad2/3 to degrade p-Smad2/3 [25,26,43]. On the basis of these studies, we determined that the stability of p-Smad2/3 was important for the transduction of TGF-β signaling and that ubiquitination was the essential factor for regulating protein stability. Moreover, numerous studies have revealed that ubiquitination is associated with the development of liver fibrosis. Particularly, in this study, we detected that the reduction of the protein level of p-Smad2/3 could be blocked by treating with proteasome inhibitor MG132, suggesting that the regulatory function of LRRC1 on p-Smad2/3 may relate to the ubiquitination. As mentioned above, the E3 ubiquitin ligase NEDD4L has been identified as primarily targeting p-Smad2/3 for degradation. Similarly, in our study, NEDD4L-mediated ubiquitination of p-Smad2/3 increased after LRRC1 was knocked down, indicating that LRRC1 affects the interaction between NEDD4L and p-Smad2/3, thereby promoting the stability of p-Smad2/3.

Although TGF-β1 signaling is considered a significant contributor to the progression of liver fibrosis, the development of antifibrotic therapies based on inhibiting TGF-β1 signaling is slow [44]. TGF-β1 receptor inhibitors, which have been studied most often, may not be appropriate options for the treatment of liver fibrosis due to the widespread presence of TGF-β1 receptors in diverse cell types and the pleiotropic functions involved [45,46]. However, recent studies have demonstrated that adeno-associated virus (AAV) vectors are promising and leading therapeutic gene delivery platforms for the treatment of human diseases [47,48,49,50]. In our study, mouse models of CCl_4_-induced liver fibrosis were injected with AAV–shLRRC1 vector via the tail vein to inhibit LRRC1 expression, and we found that knockdown of LRRC1 in vivo effectively mitigated CCl_4_-induced liver fibrosis, as evidenced by the reduction of collagen accumulation and the expressions of fibrogenic genes and p-Smad2/3, which aligns with our results in vitro. Furthermore, the hepatic Hyp content and the serum AST, ALT, and AKP levels also significantly decreased when LRRC1 was inhibited. These collective findings provide compelling evidence that the downregulation of LRRC1 expression impedes the development of liver fibrosis. Liver fibrosis is a common pathological phenomenon involving persistent liver damage with various etiologies, including HBV infection, HCV infection, and alcohol abuse. Despite variations in the specific mechanisms that trigger liver fibrosis, all of these mechanisms include common processes, such as hepatocyte injury, HSC activation, and ECM remodeling. In our current study, we demonstrated that LRRC1 regulates HSC activation and collagen expression and that the inhibition of LRRC1 in the CCl_4_-induced fibrotic mice could alleviate liver injury and liver fibrogenesis. Therefore, we propose that the regulatory mechanism by which LRRC1 promotes liver fibrosis is a pathway shared across various etiologies.

Inevitably, there are several limitations in our research. First, according to the results of the IHC analysis of human and murine fibrotic liver tissues compared to that in normal liver tissues, the expression of LRRC1, in addition to being upregulated in HSCs, was also upregulated in hepatocytes. Due to the abundance of hepatocytes in liver tissue and the regulatory role of LRRC1 in epithelial cell polarity, the weak and moderate positive expressions of LRRC1 in injured hepatocytes cannot be ignored. Moreover, the observed reduction in the serum levels of ALT, AST, and AKP after LRRC1 inhibition through AAV–shLRRC1 indicated that LRRC1 knockdown may have a beneficial effect on liver injury. Therefore, we speculated that LRRC1 may play regulatory roles not only in HSCs but also in hepatocytes, as well as in signaling crosstalk between HSCs and hepatocytes to promote liver fibrosis. However, this hypothesis still needs to be further elucidated, which is also an indispensable direction for our future research. For example, the human 3D liver system, which comprises various cell types, is considered a preferred model for further investigation of the cell-specific biological functions and mechanisms of LRRC1 in the liver. Second, research on the TGF-β1/LRRC1 axis has not yet been completed, and additional investigations are required to elucidate the downstream signaling pathways activated by LRRC1. The specific mechanisms through which LRRC1 influences the interaction between the E3 ubiquitin ligase NEDD4L and p-Smad2/3 also need to be further explored. Third, although we successfully demonstrated the regulatory role of LRRC1 in liver fibrosis both in vitro and in vivo, further investigations employing LRRC1 knockout mouse models are necessary to comprehensively explore the biological significance of the inducible expression of LRRC1.

In summary, our current study firstly demonstrated the function and underlying mechanisms of LRRC1 in liver fibrosis. Our research revealed that LRRC1 was highly expressed in human and murine liver fibrotic tissues and positively correlated with the severity of liver fibrosis. In particular, our data also indicated that LRRC1 promoted the activation of HSCs and liver fibrogenesis in vitro and in vivo. A mechanistic study elucidated that the regulatory effect of LRRC1 on liver fibrosis is due to its ability to promote the stability of p-Smad2/3. Thus, our findings contribute to the current knowledge on the mechanistic underpinnings of liver fibrosis and provide a reference for future functional studies on LRRC1.

## 4. Materials and Methods

### 4.1. Cell Line and Culture

The human hepatic stellate cell line LX-2 was purchased from the BeNa Culture Collection (Beijing, China). Cells were grown in high-glucose DMEM with the addition of 10% FBS, 100 U/mL of penicillin, and 100 μg/mL of streptomycin. The incubation took place at 37 °C in a 5% CO_2_ atmosphere with humidity.

### 4.2. Isolation of Primary HSCs

Primary rat HSCs were isolated through density gradient centrifugation [51,52]. SD rat livers were perfused in situ, digested in buffer containing collagenase IV (Solarbio, Beijing, China) and pronase E (Solarbio, Beijing, China), fragmented under sterile conditions, and subsequently immersed in a collagenase solution for additional digestion. To obtain dispersed cells, the digest solution was incubated in a thermostatic shaker (37 °C, 200 rpm) for 30 min. Afterward, the cell suspension was filtered using a 200-mesh cell strainer and then separated through centrifugation on a density gradient of Nycodenz (18%, 12%, and 8.2% Nycodenz).

### 4.3. Plasmids and Reagents

The LRRC1-pcDNA3.1-EGFP and pcDNA3.1-EGFP were purchased from FENGHUISHENGWU (Fenghui Biotechnology Co., Ltd., Changsha, China). The pAAV-U6-sgRNA-CMV-EGFP plasmid was kindly provided by associate Professor Jun Huang, Zhengzhou University. The shRNA targeting LRRC1 (pGPU6-GFP-Neo-LRRC1-Homo611) and its negative control shRNA were designed by GenePharma (Shanghai, China). The siRNAs targeting NEDD4L were synthesized by TsingKe Biotech (Beijing, China). TGF-β1 was purchased from PeproTech (PeproTech (Suzhou) Co., Ltd., Suzhou, China), and MG132 was purchased from Topscience (Shanghai, China).

The cells were transfected with plasmids or siRNAs by utilizing Lipo8000 (Beyotime, Shanghai, China) as per the guidelines provided by the manufacturer.

pAAV-U6-shLRRC1-CMV-EGFP was constructed using Spel and XbaI enzymes (Yishen, Shanghai, China). The extracted plasmid was obtained using an EndoFree Plasmid Mini Kit (Kangwei Century Biotechnology Co., Ltd., Beijing, China). All of the plasmids were verified through sequencing analysis (Sangon Bioengineering, Shanghai, China). The detailed primers used for constructing the plasmids are shown in Appendix A.

### 4.4. Animal Models and Treatments

The animals were kept in a controlled environment with a 12 h cycle of light and darkness. All animal experiments were approved by the Experimental Animal Ethics Committee of Zhengzhou University (ZZUIRB 2023-267) and conducted following the ethical guidelines of the Henan Province Experimental Animal Management Committee and the National Institutes of Health Guide for the Care and Use of Laboratory Animals.

CCl_4_-induced rat model of liver fibrosis: Male SD rats (8–10 weeks old, weighing 200 ± 20 g) were obtained from SLAC Laboratory Animal Co., Ltd. (SLAC Laboratory Animal Co., Ltd., Changsha, China) and subcutaneously injected with either a 40% CCl_4_ soybean oil solution (2 mL/kg body weight) or an equal volume of soybean oil twice a week for 6 weeks (*n* = 8 per group).

BDL-induced mouse model of liver fibrosis: Male C57BL/6 mice (6–8 weeks old, weighing 20 ± 2 g) were obtained from SLAC Laboratory Animal Co., Ltd. (SLAC Laboratory Animal Co., Ltd., Changsha, China). The mice underwent double ligation of the bile duct or sham surgery under general anesthesia (isoflurane, 0.5 L/min, inhalation anesthesia). On day 14, following BDL surgery, the mice were sacrificed (*n* = 8 per group).

CCl_4_-induced mouse model of liver fibrosis and treatments: Male C57BL/6 mice (6–8 weeks old, weighing 20 ± 2 g) were obtained from Changsheng Biotechnology Co., Ltd. (Changsheng Biotechnology Co., Ltd., Benxi, China). After adaptive feeding, the mice were randomly divided into 3 groups. In the (i) CCl_4_ + AAV–shLRRC1 group and (ii) CCl_4_ + AAV–empty group, mice were intraperitoneally injected with 25% CCl_4_ diluted in soybean oil (2 mL/kg body weight) twice a week. After two weeks, the mice were injected with the pAAV-U6-shLRRC1-CMV-EGFP or pAAV-U6-sgRNA-CMV-EGFP vector (diluted in phosphate buffer solution (5 µg/mL, 2 mL/mouse)) via the tail vein. In the (iii) Oil+AAV–empty group, mice were treated with an equal volume of soybean oil and injected with pAAV-U6-sgRNA-CMV-EGFP via the tail vein. Mice in all groups were injected with CCl_4_ or soybean oil for another 4 weeks. At the end of the experimental period, all mice were euthanized, and the blood samples and liver tissues were gathered for additional experiments.

### 4.5. Human Liver Specimens

A human liver tissue chip (Zhongke Guanghua Intelligent Biotechnology Co., Xi’an, China) was utilized in this study. Patients included in this study were 52 to 76 years old. Normal liver tissues were collected from 10 healthy subjects with ages ranging from 21 to 56 years old.

### 4.6. HE, Sirius Red, and IHC Staining

Paraffin-embedded liver tissue samples were cut into 4 µm thick slices and stained using hematoxylin and eosin (HE), Sirius Red, and IHC techniques following established protocols. The positive area of Sirius Red staining was measured using Image-Pro-Plus software (version 7.1). Fibrosis levels were assessed using the Ishak scoring system. For IHC staining, liver sections were incubated with antibodies against LRRC1 (1:200; Sanying Biotechnology Co., Ltd., Wuhan, China), α-SMA (1:4000; Sanying Biotechnology Co., Ltd., Wuhan, China), and p-Smad2/3 (1:250; Boosen Biotechnology Co., Ltd., Beijing, China). We randomly selected 3 visual fields from each slice to analyze the area and intensity of the positively stained region. The positive area (X values) was calculated using Image-Pro-Plus software. The intensity score (Y values) was based on the following guidelines: 0 indicating negative staining; 1 indicating weak staining; 2 indicating moderate staining; and 3 indicating strong staining. The IHC scores for LRRC1, α-SMA, and p-Smad2/3 were obtained by multiplying the X and Y values.

### 4.7. RNA Extraction and Quantitative Real-Time PCR

Total RNA was isolated from liver tissues and LX-2 cells using a total RNA kit (Beibei Biotechnology, Zhengzhou, China). A BeyoRT™II First Strand cDNA Synthesis Kit (Beyotime, Shanghai, China) was used to reverse transcribe the RNA (1–5 μg). SYBR Green qPCR mix (Beyotime, Shanghai, China) and a QuantStudio5 system (ABI, Thermo Fisher Scientific, Waltham, MA, USA) were utilized for qRT-PCR. The cDNA was amplified using the given procedure: 50 °C for 2 min, 95 °C for 2 min, 40 cycles of 95 °C for 15 s, and 60 °C for 30 s. The mRNA expression level of the desired genes was standardized against GAPDH using the 2^−ΔΔCt^ method. Primer sequences are listed in Appendix A.

### 4.8. Western Blot and Immunoprecipitation

Protein samples were obtained by lysing cells or liver tissues with RIPA lysis buffer (Beyotime, Shanghai, China). For Western blot analysis, equivalent amounts of protein were separated via 10% SDS-PAGE and then transferred onto nitrocellulose membranes (Pall (China) Co., Ltd., Beijing, China). The membranes were blocked with 5% skim milk for 1 h and incubated overnight with diluted primary antibodies against α-SMA (1:2000; Proteintech, Rosemont, IL, USA), Col I (1:1000; Proteintech), Col III (1:1000; Proteintech), LRRC1 (1:800; Proteintech), TGFβR I (1:1000; Bioss, Woburn, MA, USA), TGFβR II (1:1000; HUABIO, Huaan Biotechnology Co., Ltd., Hangzhou, China), p-TGFβR II (1:1000; HUABIO), Smad2/3 (1:1000; HUABIO), p-Smad2/3 (Thr8) (1:1000; Bioss), Smad7 (1:1000; HUABIO), GFP (1:2000; Proteintech), and NEDD4L (1:1000; Proteintech). After incubation with the secondary antibody (1:2000, Thermo Fisher Scientific (China) Co., Ltd., Shanghai, China) for 1 h, the target bands were visualized using a chemiluminescence imager (C300, Azure Biosystem Inc., Dublin, OH, USA).

For immunoprecipitation (IP) assay, cell lysates were combined with 4 μg of Ubiquitin antibody (Proteintech) and 30 μL of prewashed protein A/G conjugated agarose beads (Beyotime, Shanghai, China) overnight at 4 °C. The mixture was washed using ice-cold IP lysis buffer (Beyotime, Shanghai, China) five times, resuspended in 5 protein loading buffer (Solarbio, Beijing, China), and boiled for 10 min. The following steps were the same as that for the Western Blot.

### 4.9. MTT and Transwell Assays

LX-2 cells were transfected with the indicated vectors for 24 h and collected for further experiments. For the MTT assay, cells were seeded into 96-well plates (2103 each well) with 6 replicates. At every designated time interval, cells were treated with 20 microliters of MTT solution (500 ng/mL) and incubated for 4 h. Subsequently, the supernatant was substituted with 150 microliters of DMSO, followed by measuring the absorbance at 570 nm and 450 nm utilizing a microplate reader (Thermo Fisher, USA). The cell vitality was calculated as OD570 minus OD450.

For the Transwell assay, cells (2106 cells/mL) were suspended in serum-free high-glucose DMEM. The upper chamber of the Transwell insert (Corning, NY, USA) was filled with 200 µL of the cell suspension, either with or without TGF-β1 (5 ng/mL). Then, 600 µL of 10% FBS high-glucose DMEM was added to the lower chamber. After an incubation period of 46 h, non-migrating cells were eliminated using cotton swabs, while the migrated cells were fixed with 4% paraformaldehyde for 30 min and subsequently stained with 0.1% crystal violet for 30 min. The stained cells were imaged using an upright microscope at a magnification of 100× and analyzed using the software GraphPad Prism 9.0.

### 4.10. Cell Cycle Analysis

The distribution of the cell cycle was evaluated using the Cell Cycle and Apoptosis Analysis Kit (Beyotime, Shanghai, China). Briefly, LX-2 cells were transfected with the specified vectors for 48 h and then fixed with 70% cold ethanol overnight. Next, the immobilized cells were resuspended in 500 microliters of propidium iodide staining solution and incubated in darkness for 30 min. Subsequently, a flow cytometer (BD Biosciences, Franklin, TN, USA) was utilized to determine the results, which were further analyzed using the ModFit software (version 5.1) package (Verity Software House Company, Topsham, ME, USA).

### 4.11. Hydroxyproline Level and Liver Function Test

The hydroxyproline level in the liver tissues was analyzed via a hydroxyproline assay kit from Nanjing Jiancheng Bioengineering Institute (Nanjing, China). The serum levels of AKP, ALT, and AST were measured separately using the corresponding assay kits (Nanjing Jiancheng Bioengineering Institute, Nanjing, China).

### 4.12. Statistical Analysis

Data analyses were conducted utilizing SPSS software (version 25.0; IBM, Armonk, NY, USA) and GraphPad Prism software (version 9; San Diego, CA, USA). Student’s t test or the Mann–Whitney U test were used to compare values between two groups. The one-way ANOVA followed by Dunn’s multiple comparison test or the Mann–Whitney U test were used to compare three or more groups. The data are presented as the mean ± standard deviation (SD) from three biological replicates. Differences were considered statistically significant at *p* < 0.05.

## Figures and Tables

**Figure 1 ijms-25-02735-f001:**
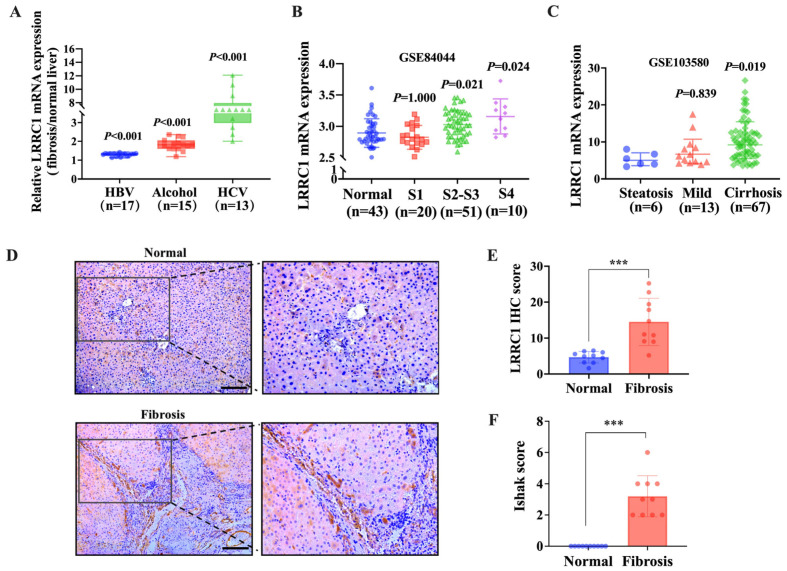
LRRC1 is highly expressed in fibrotic human livers. (**A**) The mRNA levels of LRRC1 in liver tissues from patients with HBV infection (GSE38941), alcoholic hepatitis (GSE28619), and chronic hepatitis C virus (HCV) (GSE6764). The data are shown as the ratio of the individual values for fibrosis samples to the mean values for normal control samples. (**B**,**C**) The mRNA expression of LRRC1 at different stages of liver fibrosis induced by HBV infection (B; GSE84044) and alcohol abuse (C; GSE103580). (**D**) Representative images of IHC staining for LRRC1 in normal and clinical fibrotic liver tissues (*n* = 10; scale bars, 200 µm). (**E**,**F**) Statistical analysis of the LRRC1 IHC score (**E**) and Ishak score (**F**). The data were analyzed using the Mann–Whitney U test and are presented as means ± SDs. *** *p* < 0.001 versus the normal group. LRRC1, leucine-rich repeat containing 1; HBV, hepatitis B virus; HCV, hepatitis C virus; IHC, immunohistochemistry.

**Figure 2 ijms-25-02735-f002:**
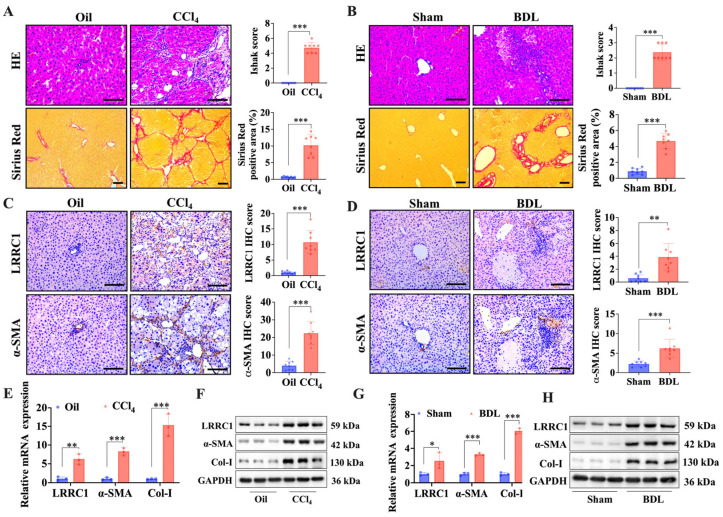
LRRC1 is upregulated in experimental murine fibrotic liver tissues. (**A**,**B**) Representative images (**left panel**) and quantification (**right panel**) of Ishak scores and Sirius-Red-positive area for collagen (HE, *n* = 8; scale bars, 200 µm; Sirius Red, *n* = 8; scale bars, 100 µm) in the indicated groups of CCl_4_-induced liver fibrosis rat model (**A**) and BDL-induced liver fibrosis mouse model (**B**). (**C**,**D**) Representative images (**left panel**) and quantification (**right panel**) of IHC staining for LRRC1 and a-SMA (*n* = 8; scale bars, 200 µm) in the indicated groups of CCl_4_-induced liver fibrosis rat model (**C**) and a BDL-induced liver fibrosis mouse model (**D**). (**E**,**F**) qRT-PCR (**E**) and Western blot analysis (**F**) of LRRC1, α-SMA, and Col-I in a CCl_4_-induced liver fibrosis rat model (*n* = 3). (**G**,**H**) qRT-PCR analysis (**G**) and Western blot analysis (**H**) of LRRC1, α-SMA, and Col-I in a BDL-induced liver fibrosis mouse model (*n* = 3). The data in (**A**–**D**) were analyzed using the Mann–Whitney U test, and the data in (**E**,**G**) were analyzed using Student’s *t* test. The data are presented as means ± SDs. * *p* < 0.05, ** *p* < 0.01, *** *p* < 0.001 versus the Oil or sham group. CCl_4_, carbon tetrachloride; BDL, bile duct ligation; HE, hematoxylin and eosin; α-SMA, α-smooth muscle actin; Col-I, type I collagen.

**Figure 3 ijms-25-02735-f003:**
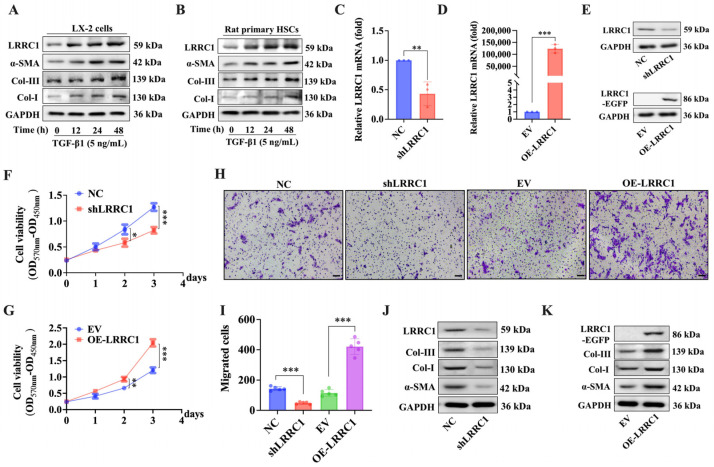
LRRC1 promotes the activation of HSCs. (**A**,**B**) Western blot analysis of LRRC1, α-SMA, Col-I, and Col-III expression in LX-2 cells (**A**) and pHSCs (**B**) after treatment with 5 ng/mL TGF-β1 for 0, 12, 24, or 48 h. The mRNA (**C**,**D**) and protein (**E**) levels of LRRC1 in LX-2 cells transfected with LRRC1-knockdown (shLRRC1) or LRRC1-overexpressing (OE-LRRC1) plasmids or the corresponding control vectors. (**F**,**G**) Growth curves showing the viability of LX-2 cells after transfection with shLRRC1, OE-LRRC1, or the corresponding control vectors. (**H**,**I**) Representative images (**H**) and quantification (**I**) of migrated cells from LRRC1-knockdown or LRRC1-overexpressing LX-2 cells and the corresponding control cells. Scale bars, 50 µm. (**J**,**K**) Western blot analysis of the expression of fibrogenic proteins (α-SMA, Col-I, and Col-III) and LRRC1 in LRRC1-knockdown or LRRC1-overexpressing LX-2 cells. The data in (**C**–**D**) were analyzed using the Mann–Whitney U test, and the data in (**F**,**G**,**I**) were analyzed using Student’s **t** test. The data are presented as means ± SDs. * *p* < 0.05, ** *p* < 0.01, *** *p* < 0.001 versus the corresponding control cells. Col-III, type III collagen; GAPDH, glyceraldehyde-3-phosphate dehydrogenase; MTT, methyl thiazolyl tetrazolium.

**Figure 4 ijms-25-02735-f004:**
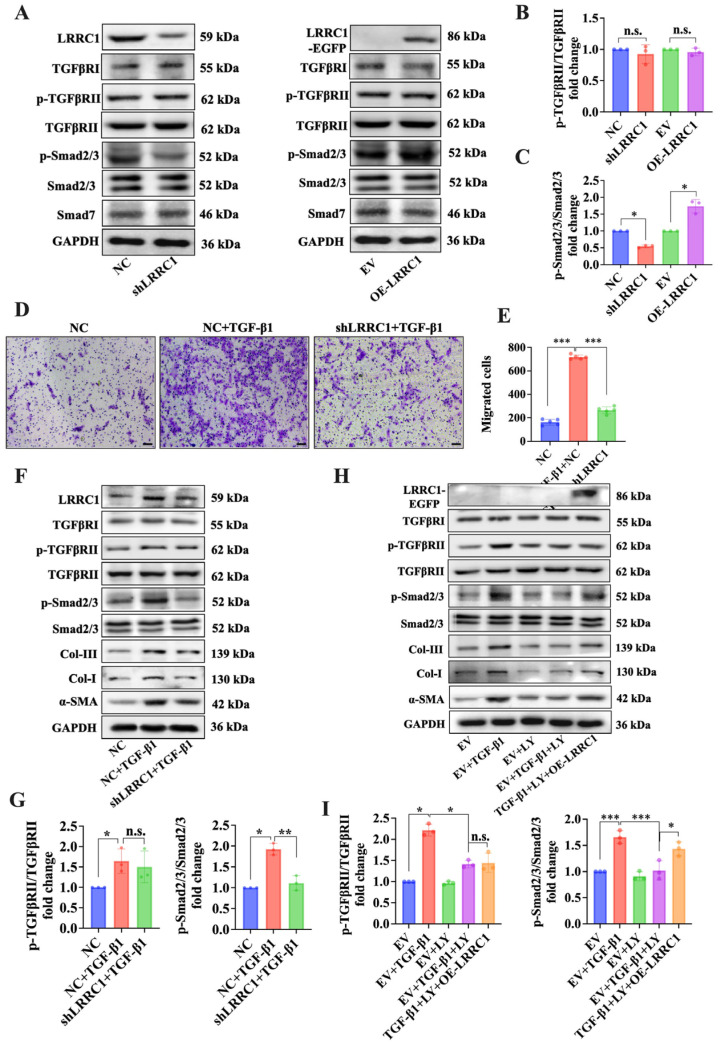
LRRC1 regulates HSC activation via the TGF-β1/Smad pathway. (**A**) Western blot analysis of the expression of TGFβRI, p-TGFβRII, TGFβRII, p-Smad2/3, Smad2/3, and Smad7 in LRRC1-knockdown or LRRC1-overexpressing LX-2 cells. (**B**,**C**) Densitometric quantitation of p-TGFβRII and p-Smad2/3 levels normalized to total TGFβRII (**B**) and total Smad2/3 (**C**) levels. (**D**,**E**) Representative images and quantification of migrated cells stimulated with TGF-β1 (5 ng/mL) after transfection with shLRRC1 or NC vectors. Scale bars, 50 µm. (**F**) Western blot analysis of the expression of TGFβRI, p-TGFβRII, TGFβRII, p-Smad2/3, Smad2/3, α-SMA, Col-I, and Col-III in LX-2 cells stimulated with TGF-β1 (5 ng/mL) after transfection with the shLRRC1 or NC vector. (**G**) Densitometric quantitation of p-TGFβRII and p-Smad2/3 levels normalized to total TGFβRII and total Smad2/3 levels. (**H**) Effect of LRRC1 overexpression on the protein expression of TGFβRI, p-TGFβRII, TGFβRII, p-Smad2/3, Smad2/3, α-SMA, Col-I, and Col-III in TGF-β1-stimulated LX-2 cells after treatment with the inhibitor LY2109761 (20 µM). (**I**) Densitometric quantitation of p-TGFβRII and p-Smad2/3 levels normalized to total TGFβRII and total Smad2/3 levels. The data in (**B**,**C**,**G**,**I**) were analyzed via one-way ANOVA followed by Dunn’s multiple comparison test. The data are presented as means ± SDs. n.s., not significant; * *p* < 0.05, ** *p* < 0.01, *** *p* < 0.001 versus the NC group, EV group, TGF-β1 group, or TGF-β1+LY2109761 group.

**Figure 5 ijms-25-02735-f005:**
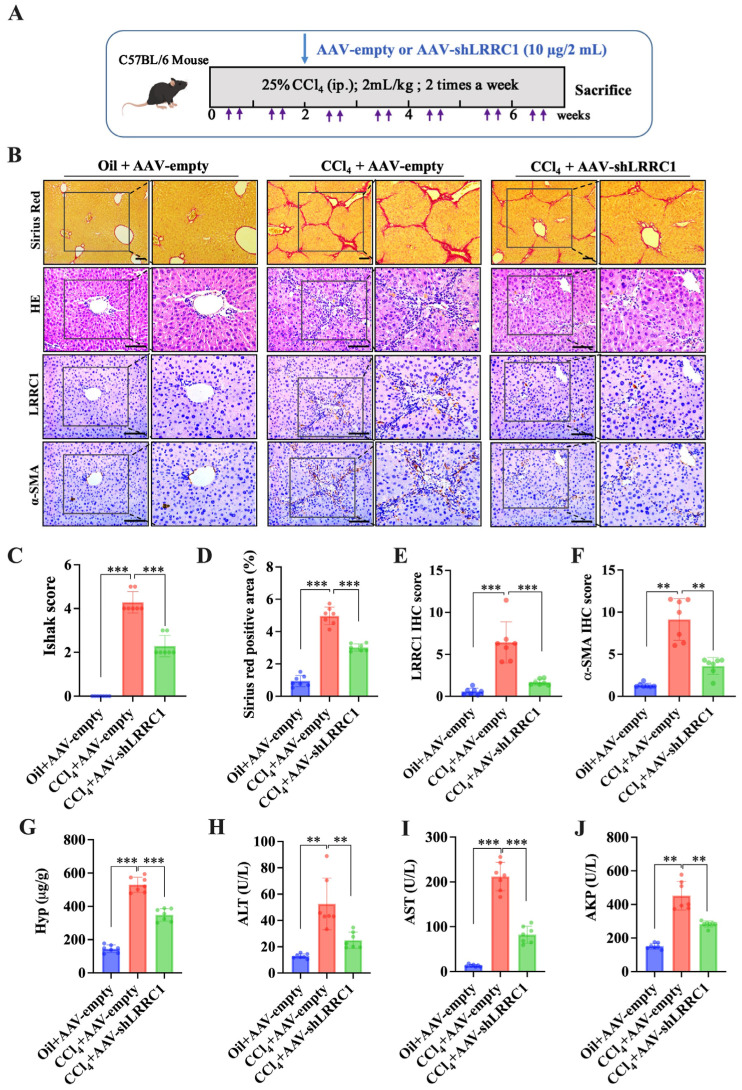
Knockdown of LRRC1 alleviates CCl_4_-induced liver fibrosis in mice. (**A**) Schematic of the experimental design for the knockdown of LRRC1 expression in the CCl_4_-induced mouse model. (**B**) Representative images of Sirius Red (scale bars, 100 µm), HE (scale bars, 200 µm), and IHC staining (scale bars, 200 µm) for LRRC1 and α-SMA in liver tissues from CCl_4_-induced model mice injected with the AAV–shLRRC1 vector or AAV–empty vector (*n* = 7). (**C**–**F**) Quantification of the Ishak score (**C**), Sirius-Red-positive area (**D**), IHC score (**E**), and α-SMA IHC score (**F**) in a CCl_4_-induced liver fibrosis mouse model generated through the injection of the AAV–shLRRC1 vector or AAV–empty vector (*n* = 7). (**G**–**J**) Hepatic hydroxyproline levels (**G**) and serum ALT (**H**), AST (**I**), and AKP (**J**) levels in CCl_4_-induced liver fibrosis model mice after the injection of the AAV–shLRRC1 vector or AAV–empty vector (*n* = 7). The data in (**C**–**J**) were analyzed using the Mann–Whitney U test, and all the data are presented as means ± SDs. ** *p* < 0.01, *** *p* < 0.001 versus the CCl_4_ + AAV–empty group.

**Figure 6 ijms-25-02735-f006:**
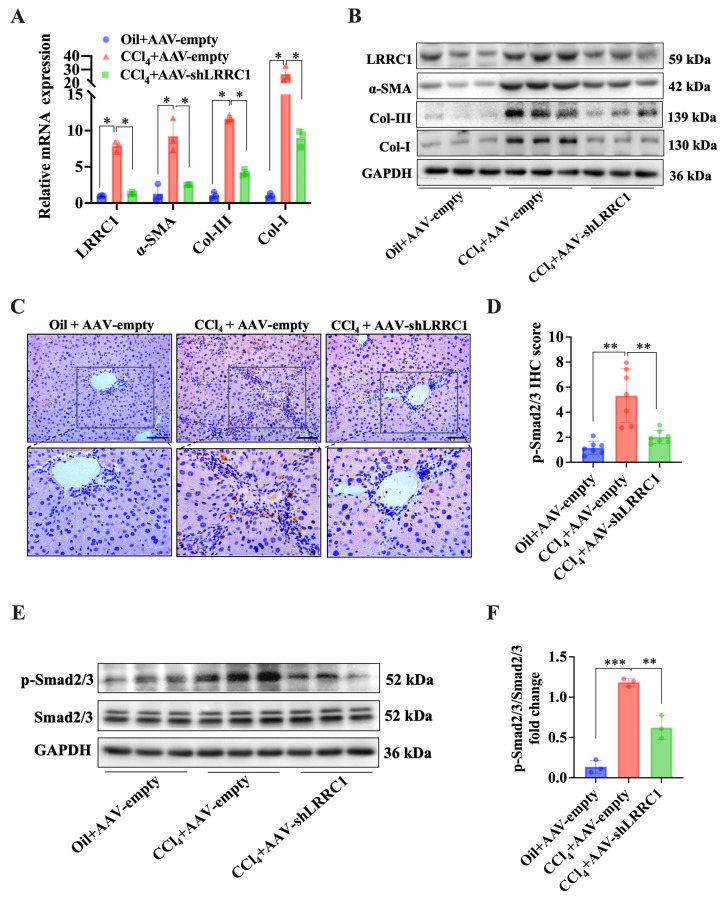
LRRC1 regulates p-Smad2/3 expression in vivo. (**A**,**B**) qRT-PCR analysis (**A**) and Western blot analysis (**B**) of the expression of LRRC1, α-SMA, Col-I, and Col-III in the liver tissues of CCl_4_-induced fibrosis model mice injected with the AAV–shLRRC1 vector or AAV–empty vector (*n* = 3). Representative images (**C**) and quantitation of IHC staining (**D**) for p-Smad2/3 in the liver tissues of CCl_4_-induced fibrosis model mice injected with the AAV–shLRRC1 vector or AAV–empty vector (*n* = 7). Scale bars, 200 µm. (**E**) Western blot analysis of p-Smad2/3 and Smad2/3 in liver tissues from CCl4-induced fibrosis model mice injected with the AAV–shLRRC1 vector or AAV–empty vector (*n* = 3). (**F**) Densitometric quantitation of p-Smad2/3 levels normalized to total Smad2/3 levels. The data in (**A**,**D**,**F**) were analyzed using the Mann–Whitney U test. The data are presented as means ± SDs. * *p* < 0.05, ** *p* < 0.01, *** *p* < 0.001 versus the CCl_4_ + AAV–empty group.

**Figure 7 ijms-25-02735-f007:**
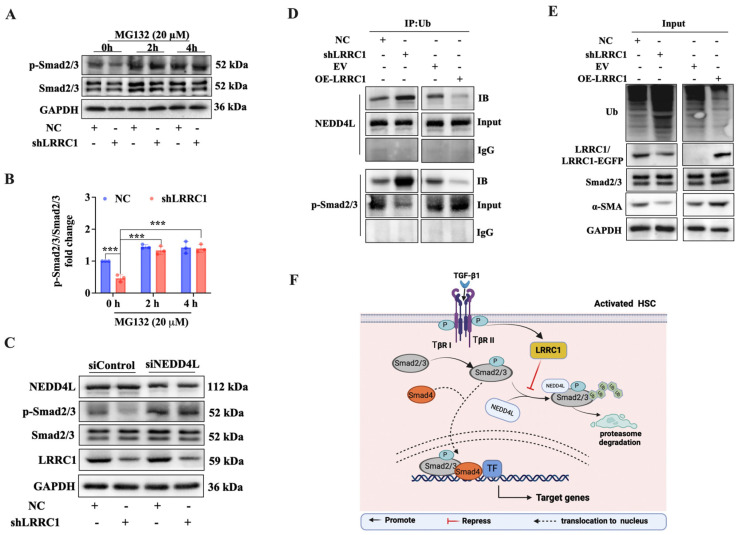
LRRC1 inhibits p–Smad2/3 proteasomal degradation. (**A**) Western blot analysis of the expression of p-Smad2/3 and Smad2/3 in LRRC1–knockdown LX–2 cells (transfected with shLRRC1 or NC vectors for 48 h) after treatment with MG132 (20 µM) at different time points. (**B**) Densitometric quantitation of p-Smad2/3 levels normalized to total Smad2/3 levels. (**C**) Effects of LRRC1 knockdown and siNEDD4L transfection on the expression of NEDD4L, p–Smad2/3, Smad2/3, and LRRC1 in LX–2 cells. (**D**,**E**) IP analysis of the interaction between ubiquitin and NEDD4L or p–Smad2/3 in LRRC1-knockdown or LRRC1-overexpressing LX-2 cells. (**F**) Working model illustrating how LRRC1 regulates liver fibrosis. In TGF-β1-stimulated HSCs, upregulated LRRC1 inhibits the interaction between NEDD4L and p–Smad2/3, resulting in a reduction in the ubiquitination-mediated degradation of p–Smad2/3, thereby facilitating TGF–β1 signal transduction to enhance HSC activation and collagen deposition, ultimately promoting the progression of liver fibrosis. The data in B were analyzed using Student’s t test and are presented as means ± SDs. *** *p* < 0.001 versus the NC group.

## Data Availability

Publicly available datasets were analyzed in this study. These data can be found here: https://www.ncbi.nlm.nih.gov/GSE38941, GSE28619, GSE6764, GSE84044, GSE103580.

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
