# Peer review of "The Upregulation of Leucine-Rich Repeat Containing 1 Expression Activates Hepatic Stellate Cells and Promotes Liver Fibrosis by Stabilizing Phosphorylated Smad2/3"

_ijms, 2024, doi:10.3390/ijms25052735_

Round 1
Reviewer 1 Report
Comments and Suggestions for Authors
The paper by Wang, et al. describes experiments testing the role of LRRC1 in hepatic fibrosis, using a well-established model (CCl4) and in vitro studies. The authors are to be commended for putting together a well-written manuscript. Additionally, each experiment is well conductetd and well described and all conclusions are supported by their data. Their results would be of great interest to researchers studying the underlying mechanisms of fibrosis and may point to potential therapies in the future. I have no concerns whatsoever as to any issues that would preclude publication in its current form.
Author Response
We sincerely appreciate the time and effort you dedicated to providing feedback on our manuscript. Your positive comments on our research have further fueled our enthusiasm for the in-depth study. We would like to extend our gratitude once more for your diligent work.
Reviewer 2 Report
Comments and Suggestions for Authors
The authors have previously shown by transcriptomic studies that LRRC1 is upregulated in CCL4-induced liver fibrosis in rats. In this study, they investigate the regulatory role of LRRC1 in LX-2 cells and demonstrate its influence on proliferation, migration and fibrogenic gene expression. In vivo experiments show that inhibition of LRRC1 restricts CCL4-induced activation of stellate cells and collagen formation in mouse livers. Based on experiments in LX-2 cells, the authors propose a mechanism involving inhibition of ubiquitin-mediated degradation of phosphorylated (Smad) 2/3 (p-Smad2/3), which activates the TGF-β1/Smad signalling pathway and induces liver fibrosis. They also present evidence for the regulation of p-Smad2/3 expression by LRRC1 in mice and the inhibition of proteasomal degradation of p-Smad2/3 in LX-2 cells. The experiments are well performed and the experimental evidence presented supports to some extent the mechanism outlined, although alternative mechanisms may also explain the data, which come from a variety of sources including LX-2 cells, HSCs and mouse livers.
Based on the results obtained in different cell systems and in mice, an overall mechanism is proposed (see Fig. 7), which is unclear to the reviewer. In the reviewers opinion the figure has to be entirely changed and corrected. Thus, the proposed mechanism needs to be explained in much more detail in the legend. The arrows are not only used for promotion but also translocation to the nucleus, interacts inhibition or interactions between NEDD4L and pSmad2/3. This leads to increased degradation of the pSmad2/3 but the complex itself does not repress ubiquitination and proteasome degradation in the manner illustrated. The right arm of the downstream pSmad2/3 pathway should be deleted as they are not independent pathways OR should not both lead to increased pSmad2/3 in the nucleus. The figure illustrated events in the HSCs by activation of target genes which includes collagen production. But due to the illustrations on the right side of the scheme, one gets the impression that the TGFBR action is within the hepatocytes which they are not.
It is difficult to discern a significant effect of NEDD4L knockdown at the protein level in the gel (7C): However, there is an obvious effect on the mobility of the protein in the control sample compared to the NEDD4L knockdown sample, possibly due to effects on phospho conjugation. It would be good to check the effect of knockdown at the mRNA level of NEDD4L. The regulation of p-smad2/3 might have other explanations.
In addition, the explanatory Figure 7F needs to be larger in order to be interpretable. It must explain which cell type is concerned and the mechanisms of HSC activation much better in higher detail. The P:s and kinases involved in this regulation would need to be specified.
The increase in LRRC is also as pointed out by the authors observed in hepatocytes and the use of a human 3D liver system containing different cell types could better identify the specific cells, mediators and mechanisms of importance. These cell-specific aspects need thus to be discussed in more detail.
LRRC1 has not yet been associated with liver fibrosis. The broad effect of LRRC1 is based on the interaction with PDZ domain-containing proteins such as DLG1 CASK, MPP7 and SNX27 and is thus involved in epithelial tissue homeostasis and tumour growth. Functionally, LRRC1 has been associated with metastatic progression in a number of malignancies, including breast and liver cancer. The action of LRRC1 on several different intracellular signalling transduction systems makes the mechanism for its action in different cellular systems being very different. The authors propose LRRC1 as a promising target for antifibrotic therapies, which may be a somewhat optimistic conclusion given the very complex mode of action of LRRC1 including very important endogenous and cancer promotion effect is other types of cells.
Specific comments
- The authors are urged to explain the migration assay and its results more fully and, in particular, to explain its significance in the context of fibrosis.
- The mechanisms that trigger the upregulation of LRRC1 in fibrotic livers should be elucidated, including the identification of the critical molecules and conditions.
- In Figure 1, the font size of the text parts (as well as in Figures 2 and 3) needs to be enlarged for better readability. The x-axis of Figure 1A should show the disease and not the GSE number of the cohort.
- Line 76 asks for clarification of the previous study to which the authors refer.
- In Figure 3, it needs to be clarified whether LX-2 cells or primary HSCs were used (F-G).
- In Figure 6B, the seemingly modest effect of shLRRC1 knockdown on protein levels is noted, although effects on mRNA expression are observed.
- Figure 7A raises questions about the discrepancy between TGF-β1 stimulation in previous experiments and the induction of SMAD2/3 phosphorylation by MG132 alone in this experiment. The duration of shLRRC1 treatment before T=0h should be indicated.
- Figure 7C gives reason to clarify the fact that knockdown of NEDD4L at the protein level is absent compared to its dominant effect on pSMAD2/3. The number of independent experiments performed should be indicated.
- In lines 265–267, the authors state that NEDD4L contributes to the stability of pSMAD2/3, which should be explained in more detail.
Comments on the Quality of English Language
OK
Reviewer 3 Report
Comments and Suggestions for Authors
The research article by Wang et al., entitled “The upregulation of LRRC1 expression activates hepatic stellate cells and promotes liver fibrosis by stabilizing p-Smad2/3” investigates the role of LRRC1 in the progression of liver fibrosis and the underlying mechanisms involved. The finding of the article is interesting and may be useful for understanding the mechanism of liver fibrosis and to develop therapy against it.
I have following concerns about the article.
1) The picrosirius red staining (Figure 2A, 2B and Figure 5B) should be perform under polarized light for better quantitative analysis of fibrosis and collagen.
2) In the figure-1A-1C, authors have shown that mRNA level of LRRC1 is upregulated in liver tissue from patients with hepatitis B, alcohol abuse and HCV infected. In the figure 1D-1E, authors have shown that, in the fibrotic tissue from clinical case, protein expression of LRRC1 is upregulated. The specific question is, which clinical case this fibrotic tissue belongs? Is it HBV/HCV/alcohol abuse? Ideally authors should present IHC images from all the three categories, as they have done for the mRNA expression level.
3) From the figure 1D, it seems that LRRC1 is not only expressing in the hepatic stellate cells, but it is also expressing in the hepatocyte. Although authors have mentioned this specific point in the limitation section of the discussion, but this needs more clarification. The reason for raising this point is the animal experiments with AAV-shLRRC1 vector. When AAV-shLRRC1 vector will be deliver to the mice (CCL4 induced liver fibrosis model), it will not only inhibit the LLRC1 gene expression in the stellate cells, but it can also inhibit the LLRC1 gene expression in the hepatocyte, therefore it is hard to convince from the animal experiment that specific inhibition in the stellate cells can regress the liver fibrosis.
4) Different etiologies like HCV infection, HBV infection, alcohol abuse, may follow different molecular mechanisms for promoting liver fibrosis/cirrhosis. In this study, it seems that LRRC1 is common to different causative agents. Do the authors conclude that it is a common pathway to liver fibrosis for all the three agents (HBV, HCV, and alcohol abuse). This needs more detail explanation in the discussion section of the manuscript.
